# Effects of Gonadotropin-Releasing Hormone (GnRH) and Its Analogues on the Physiological Behaviors and Hormone Content of *Tetrahymena pyriformis*

**DOI:** 10.3390/ijms20225711

**Published:** 2019-11-14

**Authors:** Eszter Lajkó, Éva Pállinger, Zsombor Kovács, Ildikó Szabó, László Kőhidai

**Affiliations:** 1Department Genetics, Cell and Immunobiology, Semmelweis University, Nagyvárad tér 4., 1089 Budapest, Hungary; lajesz@gmail.com (E.L.); pallinger.eva@med.semmelweis-univ.hu (É.P.); zsomborkovacs@freemail.hu (Z.K.); 2Research Group of Peptide Chemistry, Hungarian Academy of Sciences, Eötvös Loránd University, Pázmány Péter sétány 1/A, 1117 Budapest, Hungary

**Keywords:** GnRH-III, *Tetrahymena*, protozoan hormone, chemotaxis, chemokinesis, tracking analysis, cell proliferation

## Abstract

The unicellular *Tetrahymena* distinguishes structure-related vertebrate hormones by its chemosensory reactions. In the present work, the selectivity of hormone receptors was evaluated by analyzing the effects of various gonadotropin-releasing hormone (GnRH) analogs (GnRH-I, GnRH-III) as well as truncated (Ac-SHDWKPG-NH_2_) and dimer derivatives ([GnRH-III(C)]_2_ and [GnRH-III(CGFLG)]_2_) of GnRH-III on (i) locomotory behaviors, (ii) cell proliferation, and (iii) intracellular hormone contents of *Tetrahymena pyriformis*. The migration, intracellular hormone content, and proliferation of *Tetrahymena* were investigated by microscope-assisted tracking analysis, flow cytometry, and a CASY TT cell counter, respectively. Depending on the length of linker sequence between the two GnRH-III monomers, the GnRH-III dimers had the opposite effect on *Tetrahymena* migration. [GnRH-III(CGFLG)]_2_ dimer had a slow, serpentine-like movement, while [GnRH-III(C)]_2_ dimer had a rather linear swimming pattern. All GnRH-III derivatives significantly induced cell growth after 6 h incubation. Endogenous histamine content was uniformly enhanced by Ac-SHDWKPG-NH_2_ and GnRH-III dimers, while some differences between the hormonal activities of GnRHs were manifested in their effects on intracellular levels of serotonin and endorphin. The GnRH peptides could directly affect *Tetrahymena* migration and proliferation in a structure-dependent manner, and they could indirectly regulate these reactions by paracrine/autocrine mechanisms. Present results support the theory that recognition ability and selectivity of mammalian hormone receptors can be deduced from a phylogenetically ancient level like the unicellular *Tetrahymena*.

## 1. Introduction

In the early seventies of the last century, the presence of receptors for hormones (e.g., insulin receptor) of higher vertebrates has been demonstrated in the unicellular *Tetrahymena* [1,2,3]. Later, the presence of vertebrate hormones themselves (e.g., serotonin [4], insulin [5], histamine [6], and endorphin-1 [7]) and signal pathways (e.g., cAMP [8], calmodulin [9], and cGMP [10]) was also observed. The similarities of the receptors, hormones, and signal pathways to the vertebrate ones were demonstrated [3,5,11]. The recognition of hormones of the higher-ranked animal proved to influence (i) the chemosensory behavior of *Tetrahymena*, such as chemotaxis [12] and chemokinesis [13], (ii) cell survival and proliferation [14], as well as (iii) production and secretion of hormones, which imply paracrine/autocrine communication [3].

Chemotaxis is a directed movement of the cells along a chemical gradient either toward or away from a chemical stimulus [12], whereas chemokinesis denotes an unoriented change in swimming speed or turning frequency of movements, which refers to the windings of the swimming path [15]. The study of the chemotactic effects of chemokines proved that the chemosensitivity of *Tetrahymena* is similar to multicellular organisms [16]. Former data about the highly sensitive chemotactic responsiveness of *Tetrahymena* to peptide-type hormones and their structure-related derivatives (e.g., bradykinin and its truncated fragments [17]; oxytocin and its synthetic analogues [18]) point to the selectivity of these hormone receptors.

Effects of the exogenously applied hormones on the production [19] and the binding [20,21] of other hormones were studied in *Tetrahymena* in several previous experiments. Based on the results of the aforementioned studies, these hormones could also have regulatory or communicational functions in this low level of phylogeny by influencing the reactions of *Tetrahymena* at relatively high (micro- to nanomolar) and even at extremely low femto- to zeptomolar concentrations [22,23]. The sensitivity of the hormone receptors has been also indicated by these studies.

In our present work, the chemosensory responses of *Tetrahymena* induced by gonadotropin-releasing hormone (GnRH) as well as its natural and synthetic derivatives were investigated. GnRH was first discovered in mammals, but three major types of GnRH and three major types of G protein-coupled GnRH receptors (GnRH-R) are distributed in all seven classes of vertebrates. GnRH isoforms, which are known to be produced in the hypothalamus, are decapeptides with several highly conserved residues and have a significant function (e.g., neuroendocrine: the release of luteinizing hormone and follicle-stimulating hormone from the pituitary; paracrine/autocrine) through phylogeny [24,25]. Investigation of the evolution of GnRH revealed that GnRH signaling occurs throughout the chordates (e.g., lamprey: GnRH-III) and invertebrates (e.g., *Drosophila melanogaster* and *Caenorhabditis elegans*) as well. Invertebrate GnRH-like peptides such as corazonin and adipokinetic hormone share several conserved residues with the vertebrate GnRH (e.g., pGlu^1^, Ser^4^, Gly^6^, Trp^7^) and their receptors (orthologues of GnRH-R [26]), which indicate a very early evolutionary origin [25,27]. In protochordates, one of the most ancient GnRH forms was identified as a disulfide-linked dimer. Dimerization might increase the stability of the hormone and gives it a chance to crosslink receptors by binding each monomeric subunit of the dimer peptide to a receptor [28]. Although GnRH and its receptor has not been identified in *Tetrahymena pyriformis*, several studies suggest the presence and importance of pituitary hormone-like material at this low level of the phylogeny. A diffuse localization of molecules immunologically similar to human gonadotropins has been already reported in *Tetrahymena* [29]. It was also shown that thyroid-stimulating hormone (TSH)-mediated triiodothyronine (T3) production can be deduced to a unicellular level [30].

In our previous experiments, the structure-dependent chemotaxis of *Tetrahymena* was detected towards different GnRH peptides [31]. Mammalian GnRH-I (GnRH-I; Glp-His-Trp-Ser-Tyr-Gly-Leu-Arg-Pro-Gly-NH_2_) and the more ancient lamprey GnRH-III (GnRH-III; Glp-His-Trp-Ser-His-Asp-Trp-Lys-Pro-Gly-NH_2_) had opposite chemotactic effects (GnRH-I: weak chemoattractant; GnRH-III: chemorepellent). The length of the hormone was indicated to determine the chemotactic responsiveness of *Tetrahymena* by detecting the chemoattractant character of an N-terminal truncated fragment of GnRH-III (Ac-SHDWKPG-NH_2_). This fragment and some other N-terminal truncated GnRH-III peptides were previously analyzed in our laboratory. In spite of the importance of the conserved N-terminal sequence for receptor binding, the cellular uptake of shortened GnRH-III peptides proved to be similar to the full-length GnRH-III in HT-29 human colon and MCF-7 human breast cancer cells [32]. Although, shorter GnRH sequences are frequently produced as a result of normal proteolytic degradation by different endo- and exopeptidases (e.g., α-chymotrypsin [33], angiotensin-converting enzyme, and thimet oligopeptidase [34]). Only few preliminary works were carried out in the 1970s and 1980s with the N- or C-terminal truncated GnRH-I fragments [35,36] or hexapeptide GnRH-I analogs [37], and a wide range of binding affinities (equal to or lower than that of the GnRH) and biological responses (both agonistic and antagonistic) were shown in those studies. Wu’s group has recently shown that GnRH-(1-5), produced by thimet oligopeptidase (zinc metalloendopeptidase EC3.4.24.15), has biological activity (e.g., regulating GnRH-I synthesis, secretion, and migration of GnRH neurons), which could be different or even antagonistic compared to the parent peptide [34,38]. In our previous work, different symmetric GnRH-III dimers were synthesized via a disulfide bridge between two GnRH-III molecules containing branch in position 8 [33]. The incorporation of GFLG tetrapeptide between GnRH-III monomer and the disulfide bridge ([GnRH-III(CGFLG)]_2_) turned the chemoattractant activity of [GnRH-III(C)]_2_ dimer into a wide range of repellent activities. This study also showed that the GnRHs, besides their chemotactic effects in *Tetrahymena,* could be inducers of phosphatidylinositide 3-kinase involved in GnRH-R signaling [31].

Based on the fact that hormones of higher-ranked animals could influence many cell physiological reactions of *Tetrahymena*, the present experiments make an attempt
(i)to investigate the more detailed effects of GnRH and its derivatives in *Tetrahymena*;(ii)to find some correlation between the different cell physiological reactions of *Tetrahymena* induced by various GnRH derivatives;(iii)to evaluate the selectivity of chemosensory behaviors of *Tetrahymena* and, consequently, the selectivity of its receptors;(iv)to interpret the roles of GnRH peptides at the unicellular level of the phylogeny.

## 2. Results

In the present work, five GnRH derivatives (GnRH-I at 10^−6^ M, GnRH-III at 10^−6^ M, Ac-SHDWKPG-NH_2_ at 10^−6^ M, [GnRH-III(C)]_2_ at 10^−11^ M, and [GnRH-III(CGFLG)]_2_ at 10^−6^ M), which were found chemotactically active in our previous study [31], were investigated. The results of the assays with GnRH peptides are categorized and presented in the following way: (i) natural hormones (GnRH-I and GnRH-III), (ii) N-terminal truncated fragment (Ac-SHDWKPG-NH_2_), and (iii) symmetric GnRH-III dimers ([GnRH-III(C)]_2_ and [GnRH-III(CGFLG)]_2_). The molecular composition of the tested GnRH derivatives is shown in Table 1. The concentrations that were applied in the measurements of chemokinesis (i.e., swimming behavior) and hormone content were identical to those where the most significant chemotactic character (chemoattractant: GnRH-I, Ac-SHDWKPG-NH_2_, [GnRH-III(C)]_2_; chemorepellent: GnRH-III, [GnRH-III(CGFLG)]_2_ was observed in our previous study [31]. The different cell physiological factors of *Tetrahymena*, investigated in our present work, and the effects of the mammalian hormones, including GnRHs, on them are not completely independent from each other. In order to investigate the connection between the different cell physiological activities (e.g., chemotactic, chemokinetic, and endocrine activities) induced by GnRH derivatives, it was reasonable to apply the same concentration (found to be the most active in the chemotaxis assay) during the other measurements as well.

### 2.1. Chemokinesis Swimming Behavior

The mean swimming velocity (expressed as a percentage of the control) and the tortuosity (tort, the ratio of total distance to straight distance) were used to characterize the change in the chemokinetic activity of *Tetrahymena* induced by GnRH derivatives. The tortuosity is used to describe the windings of the swimming path. If the value of tortuosity is 1, it means that the swimming track is straight; if this value is between 1 and 10, the cell swims in a more spiral (winding) path; and if the tortuosity is greater than 10, *Tetrahymena* performs a slow, circular type of movement known as creeping [13]. The results of the chemokinetic measurements are shown in Figure 1 and Appendix A.

In the case of the untreated control cells, according to the literature [13,39], rapid, serpentine-like swimming (tort = 1.56) was observed (Figure 1b). The natural isoforms had a similar chemokinetic profile, both GnRH-I and GnRH-III caused more winding swimming paths (tort = 2.8, *F*_1,97_ = 9.04, *p* = 0.0036 and tort = 2.7, *F*_1,96_ = 7.72, *p* = 0.0065) than the control (Figure 1b) and simultaneously decreased the velocity (76.3 %, *F*_1,97_ = 10.17, *p* = 0.0019 and 78.1 %, *F*_1,96_ = 7.53, *p* = 0.0072) (Figure 1a).

The Ac-SHDWKPG-NH_2_ fragment markedly reduced the mean velocity (58.5 %, *F*_1,103_ = 29.50, *p* = 3.75 × 10^−7^) of the cells (Figure 1a); however, there was no significant difference in the windings of cell paths (tort = 1.89, *F*_1,103_ = 2.03, *p* = 0.1569) compared with the control (Figure 1b). In the case of the [GnRH-III(C)]_2_ dimer, a slight, nonsignificant increase could be detected in the level of tortuosity of the swimming tracks (tort = 2.03, *F*_1,89_ = 2.49, *p* = 0.1178) (Figure 1b). In contrast, the [GnRH-III(CGFLG)]_2_ dimer provoked the most notable tortuosity of *Tetrahymena* (tort = 3.12, *F*_1,104_ = 10.45, *p* = 0.0016) (Figure 1b) and, in parallel, decreased the swimming velocity (61.5%, *F*_1,104_ = 21.35, *p* = 1.10 × 10^−5^) (Figure 1a) in comparison with the effects of both the control and [GnRH-III(C)]_2_ dimer.

### 2.2. Hormone Content

The effects of the exogenously given GnRH peptides were studied on the intracellular T3, histamine, serotonin, epinephrine, and endorphin contents of *Tetrahymena* by flow cytometry. The evaluation of the effects of GnRHs was done by comparing mean fluorescence intensities (dimensionless geometric mean channel values) of the treated groups to the appropriate control (Table 2). In Appendix A, the mean fluorescent intensity expressed as a percentage of the untreated control can be found.

In the case of T3 and serotonin, GnRH-I proved to be neutral, while GnRH-III had a positive effect on T3 content (63.23, *F*_1,6_ = 5.76, *p* = 0.053 vs. control: 59.00) and a negative activity on serotonin content (19.65, *F*_1,6_ = 13.28, *p* = 0.0107 vs. control: 24.04). The intracellular histamine concentration was increased by only GnRH-I (24.89, *F*_1,6_ = 36.68, *p* = 9.18 × 10^−4^ vs. control: 21.02). GnRH-I and GnRH-III increased the amount of endorphin (GnRH-I: 55.78, *F*_1,6_ = 33.846, *p* = 0.0011 and GnRH-III: 61.52, *F*_1,6_ = 56.17, *p* = 2.91 × 10^−4^ vs. control: 47.89) (Table 2).

Exposure to Ac-SHDWKPG-NH_2_ slightly reduced the serotonin (21.91, *F*_1,6_ = 4.66, *p* = 0.0741 vs. control: 24.04), epinephrine (73.03, *F*_1,6_ = 40.08, *p* = 7.26 × 10^−4^ vs. control: 100.82), and endorphin (43.85, *F*_1,6_ = 5.85, *p* = 0.051 vs. control: 47.89) content of *Tetrahymena*. Intracellular histamine was significantly elevated (30.27, *F*_1,6_ = 13.20, *p* = 5.95 × 10^−4^ vs. control: 21.02) after treatment with the fragment related both to the control and to all other GnRH derivatives (Table 2). The [GnRH-III(C)]_2_ and [GnRH-III(CGFLG)]_2_ dimers increased the level of histamine ([GnRH-III(C)]_2_: 28.59, *F*_1,6_ = 22.16, *p* = 0.0033 and [GnRH-III(CGFLG)]_2_: 27.93, *F*_1,6_ = 10.00, *p* = 0.019 vs. control: 21.02) and decreased the content of epinephrine ([GnRH-III(C)]_2_: 78.44, *F*_1,6_ = 21.79, *p* = 0.0034 and [GnRH-III(CGFLG)]_2_: 71.58, *F*_1,6_ = 37.43, *p* = 8.71 × 10^−4^ vs. control: 100.82) in a similar manner. In the case of the [GnRH-III(C)]_2_ dimer, a weak stimulating effect for the endorphin level (54.60, *F*_1,6_ = 5.70, *p* = 0.054 vs. control: 47.89) and a reduced activity for serotonin content (22.01, *F*_1,6_ = 6.41, *p* = 0.044 vs. control: 24.04) were observed. The [GnRH-III(CGFLG)]_2_ dimer had a slight, but significant, negative effect on the T3 level (54.56, *F*_1,6_ = 7.70, *p* = 0.032 vs. control: 59.00) (Table 2).

### 2.3. Cell Proliferation

The growth rate of *Tetrahymena* treated with GnRH peptides was detected over the short-term (after 6 h) and long-term (after 24 h). Proliferation index represents the percent of viable cells normalized to an identical control. The more significant results of the short-term treatments are shown in Figure 2 and Appendix A, while the results of 24 h treatments can be found in Appendix A and Appendix A.

The native hormones had only a moderate effect on *Tetrahymena* proliferation. The negative effect of GnRH-I (86.7% and 88.0%, *F_7_*_,24_ = 1.33, *p* = 0.023 and 0.039 at 10^−11^ and 10^−8^ M) was more prominent after 6 h of incubation (Figure 2a), while GnRH-III could slightly increase the growth in the long-term, but only at 10^−6^ M concentration (109.1%, *F*_7,21_ = 1.036, *p* = 0.039, *p* < 0.05) (Appendix A).

Both fragments and dimers of GnRH-III showed proliferative activity at higher concentrations (Ac-SHDWKPG-NH_2_: 121.6%, *F_7_*_,22_ = 1.33, *p* = 0.03 at 10^−8^ M; [GnRH-III(C)]_2_: 119.4–128.1%, *F*_7,20_ = 1.513, *p* = 0.067 and 0.0066 at 10^−7^–10^−6^ M; [GnRH-III(CGFLG)]_2_: 121.6%, *F*_7,24_ = 1.965, *p* = 0.013 at 10^−6^ M) in the short-term (Figure 2b). After 24 h, all GnRH-III derivatives failed to have any effect on *Tetrahymena* growth (Appendix A). 

## 3. Discussion

### 3.1. Migratory Responses of Tetrahymena

Migratory reactions of *Tetrahymena* towards a ligand are considered as chemotactic or chemokinetic responses. These responses are not completely distinguished from each other: a molecule could induce chemotaxis when acting along a concentration gradient and could influence the chemokinetic behavior of *Tetrahymena* when it presents in a homogenous (nongradient-like) concentration [15]. The swimming activity of *Tetrahymena* consists of straight runs and turns. The runs provide fast and straight elements of movement, while an increasing number of turns causes more winding in the path and consequently a larger tortuosity of tracks [40,41]. By modulating these swimming elements, *Tetrahymena* accomplishes its chemotactic responses rather than by orienting and swimming toward or away from a chemical stimulus [39,40]. Thus, a chemoattractant molecule (e.g., proteose peptone) can induce straight (lower tortuosity) and relatively fast swimming by decreasing the frequency of directional changes, while in the case of a repellent ligand (e.g., citronellol), the number of turns can be increased, and, consequently, the movement of *Tetrahymena* can become more winding (higher tortuosity) and slow [41].

In this study, we analyzed how chemotactically active GnRHs could modulate the swimming behavior of our model cell and whether there is any correlation between these two migratory responses. Our former chemotaxis results have disclosed that *Tetrahymena* could widely distinguish the GnRH peptides having smaller or greater structural differences by its chemotactic reactions [31]. In general, the swimming velocity and the tortuosity were changed mostly in a reciprocal way; the movement of the cells became more rectilinear (lower tortuosity) along with the increasing speed of cells, and slow swimming was performed on a more winding track (higher tortuosity). In good agreement with the literature [41], the chemorepellent GnRH peptides (GnRH-III and [GnRH-III(CGFLG)]_2_) provoked slow, more tortuous movements of *Tetrahymena* compared to the chemoattractant GnRH derivatives (e.g., [GnRH-III(C)]_2_ dimer) (Table 3). Our unpublished results about the chemokinetic effects of other hypothalamo/hypophyseal hormones such as TSH and human chorionic gonadotropin (hCG) proved to confirm the correlation between the chemotactic and chemokinetic effects of the tested GnRH peptides. TSH and hCG, according to their well-documented chemorepellent activities [42], had negative effects on the swimming velocity and enhanced the tortuosity of the tracks.

### 3.2. Effects of GnRH Derivatives on the Hormone Contents of Tetrahymena

In order to evaluate the complex cell biological activity of GnRH peptides on *Tetrahymena*, it was reasonable to analyze the native and synthetic GnRH peptides with respect to their effects on different intracellular hormone contents.

The flow cytometric analysis illustrated that the effects of GnRH peptides were not completely specific to a given hormone; they could simultaneously modulate the intracellular level of different hormones, even in an opposite manner. All synthetic GnRH-III derivatives elevated the intracellular level of histamine and, at the same time, decreased the level of endogenous epinephrine. This means that instead of the hormone specific effect of GnRH, the direction of the hormone content response is specific. In this case caution is warranted, since it is not known whether intracellular hormone synthesis or the secretion and storage of them are influenced in the opposite direction.

The fact that the intracellular hormone content of *Tetrahymena* could be regulated by external hormonal stimuli might predict that the cells could respond to this kind of stimulation by secreting these endogenous substances. The intracellular hormones investigated in our experiment are known to have significant chemotactic [2,12] and, therefore, chemokinetic activities. This means that after their release, these hormones could react to the same cell’s (autocrine) and/or to other cell’s (paracrine) physiological activities [3]. Changes in the intracellular hormone content could be followed by their altered secretion, which might indirectly affect the migratory response of the cells to an exogenously given hormone (e.g., GnRH) by an autocrine/paracrine manner. For example, we hypothesize that the secretion of serotonin, a strong chemorepellent hormone, might modify the chemokinetic effect of the exogenously given GnRHs by provoking a slower and more winding swimming pattern. The opposite, a fast, more linear swimming, might be caused by secreting histamine or endorphin (both of them are chemoattractant).

For a more accurate evaluation of the paracrine/autocrine regulatory function of GnRH derivatives, further investigations, including detection of secreted endogenous hormones outside the cells, would be needed. To study the concentrations and activity of the hormones released from the cells, the following points should be addressed: (i) whether these hormones are secreted in a constitutive or regulated way; (ii) what ratio of the induced endogenous hormone can be secreted; (iii) the results of modulating different intracellular hormone contents might have positive or negative effects on their release; (iv) molecular interactions (e.g., enhancing or neutralizing) might exist between environmental substances and the released endogenous hormones; and (v) the optimal time frames and concentration ranges of the paracrine/autocrine hormones should be investigated. The complexity of the problem is increased further by considering the chemokinetic/chemotactic reactions of *Tetrahymena*. The constant swimming behavior of ciliates consequently destroys the gradients of secreted hormones around the cells. Therefore, it is supposed that we count more homogenous hormone level(s), which elicits more chemokinetic than chemotactic responses. Nevertheless, the “chemotaxis–chemokinesis and intracellular hormone content system” is considered to be fundamental, and our present work offers a good attempt of its evaluation. The comparison between chemotaxis, chemokinesis, and intracellular hormone content induced by GnRH derivatives is shown in Table 3. There were significant differences between the [GnRH-III(C)]_2_ and [GnRH-III(CGFLG)]_2_ dimers in their chemotactic and chemokinetic characters, which were also manifested in their effects on the hormone content of *Tetrahymena*. Overall, the most chemoattractant [GnRH-III(C)]_2_ dimer caused abigger increase in the content of chemoattractant hormones (histamine [12], endorphin [2]) than the chemorepellent [GnRH-III(CGFLG)]_2_ dimer, and it lowered the concentration of the repellent serotonin [12]. In the case of the [GnRH-III(C)]_2_ dimer, elevated levels of endogenous chemoattractant hormones might contribute to the fast and linear swimming of *Tetrahymena*, while the cells treated with the [GnRH-III(CGFLG)]_2_ dimer were characterized by an increased level of chemorepellent hormones that led to slow, winding paths. In our previous experiment, the neutral effects of TSH and hCG were shown on endogenous endorphin and serotonin concentrations [43]. These results, in parallel with their migratory character of chemorepellence [42] and a more circular swimming pattern (our unpublished results), confirmed the above-described tendency between locomotory and hormonal responses.

According to our hypothesis, in the case of the chemoattractant GnRH peptides (e.g., GnRH-I and [GnRH-III(C)]_2_), the differences observed in the intensity of their chemotactic activity (e.g., GnRH-I: weak attractant, [GnRH-III(C)]_2_: strong attractant) and in their effects on swimming behavior (GnRH-I: tortuous paths, [GnRH-III(C)]_2_: more straight swimming) could arise from their different endocrine effects.

### 3.3. Effect of GnRH Peptides on the Proliferation of Tetrahymena

Cell proliferation and survival are the most characteristic functions of free-living unicellular organisms, and signal molecules like hormones (e.g., insulin) are absolutely necessary for *Tetrahymena* to survive and multiply, even in nutrient-rich conditions [44].

In general, GnRH peptides proved to have time-dependent effects on *Tetrahymena* proliferation; their effect was manifested in the early period, while in the long-term they were insignificant. A similar time dependency was also observed with other mammalian hormones (e.g., insulin and histamine [45]). The average doubling time for *Tetrahymena* is short (~150 min). In 6 h of incubation, *Tetrahymena* divides at least twice, which is generally accepted in a cell proliferation assay on mammalian cell lines. It can be supposed that *Tetrahymena* can decompose hormones, which results in less or no effectivity in the long-term [45,46,47]. Nevertheless, the downregulation of hormone receptors can also lead to desensitization or unresponsiveness of *Tetrahymena* in the case of continuous exposure to a hormone [48].

There can be a bidirectional relationship between hormonal content and cell proliferation. The conditions of growth strongly influence the hormone synthesis (content) of *Tetrahymena*; for instance, endogenous epinephrine [49] or serotonin levels were found to be different in the stationary and logarithmic phases [50]. At the same time, the intracellular hormone concentration could be a regulator of cell division, as it was shown in the case of histamine [50]. Interestingly, all GnRH-III derivatives enhanced the cell growth, while the natural hormones showed only a slight activity. This difference correlated well with the hormonal effects of peptides. For example, the fragment and the dimers elevated the histamine content, which is an inducer of proliferation. These results provide further evidence that there can be an interaction between the proliferation and intracellular hormone content. Furthermore, *Tetrahymena* can differentiate between related hormones/peptides; however, proliferation is not a more sensitive reaction than the migratory responses.

### 3.4. Evolutionary Aspects of Tetrahymena’s Reactions to GnRH Peptides

The responses of *Tetrahymena* to the GnRH peptides indicate this protozoon is more responsive to GnRH isoforms representing more ancient levels of the phylogeny. In comparing the natural analogues, the lamprey GnRH-III seemed to be more active in modulating the endogenous hormone content. Moreover, GnRH-III was the only hormone that induced growth in the long-term, while the mammalian GnRH-I decreased proliferation. In our study, the disulfide-linked dimers ([GnRH-III(C)]_2_ and GnRH-III(CGFLG)]_2_) were the most active peptides regarding all tested cell physiological indices; however, they had opposite effects in most of the cases. The dimer form of GnRH was shown to be typical in tunicates (e.g., *Chelyosoma productum*) [28]. However, [GnRH-III(C)]_2_ and GnRH-III(CGFLG)]_2_ dimers are not native isoforms but may represent ancestral analogues. In a former study, it was also concluded that the precursor of thyroxine (e.g., diiodotyrosine), representing lower levels of hormone phylogenesis, had a more intense effect in *Tetrahymena* than the vertebrate hormones T3 and thyroxine [51].

The above-mentioned antagonistic effects of dimers might be explained by the existence of at least two GnRH-Rs or by the induction of different signaling pathways of a single receptor. In the latter case (ligand-induced selective signaling), certain GnRH variants are able to preferentially stabilize different conformations of the GnRH receptor, which generates distinct binding and signaling outputs [52]. Koch’s theory also supports the explanation of ligand-induced selective signaling. According to this theory, proteins of the unicellular plasma membrane are “not complete”, and there is a continuous change in the conformation and/or subunits of membrane proteins. Depending on the exogenous hormone, different conformations or combinations of these membrane proteins can be stabilized in order to fulfil receptor functions [53].

## 4. Conclusions

In summary, our experiment clearly showed that GnRH peptides can influence many essential cell physiological processes of *Tetrahymena*. The structural modifications of GnRH peptides resulted in different migratory activities as well as changes in cell proliferation and the levels of intracellular hormones. This shows that, at a unicellular level, membrane structures (receptors, binding sites) are present that are suitable for recognizing hormones characteristic to higher-ranked animals, or these can develop from subpatterns in the presence of hormones [53]. In addition, these data would prove the selectivity of chemoreception and adequate functioning of signaling pathways. On the other hand, the selective reactions of *Tetrahymena* were more pronounced in the case of migratory responses than changes in endogenous hormonal levels and cell proliferation. Our results confirmed that the repellent GnRH peptides provoke slow and tortuous swimming, while in the presence of an attractant, a rather linear swimming pattern was characteristic to *Tetrahymena*. The authors suppose that, besides the direct activities of GnRH peptides on migration and cell growth, they could also influence these reactions of *Tetrahymena* by paracrine/autocrine mechanisms. The results support the earlier theory that the recognition ability and selectivity of mammalian (human) hormone receptors can be deduced to a phylogenetically ancient level, such as the unicellular *Tetrahymena* [3].

## 5. Materials and Methods

### 5.1. Tested Peptides

The tested GnRH peptides were synthesized by solid-phase peptide synthesis using a mixed Boc/Fmoc strategy. The symmetric dimers were formed using air oxidation to establish a disulfide bridge between the cysteine-modified, branched GnRH-III monomers. The crude products were purified by semipreparative RP-HPLC (Knauer GmbH, Bad Homburg, Germany). For the analytical characterization, RP-HPLC (Knauer GmbH, Bad Homburg, Germany) and electrospray ionization mass spectrometry (ESI-MS; Bruker Daltonics Esquire 3000 Plus, Bremen, Germany) were applied [31,33]. All GnRH peptides are water-soluble, and the dilutions of the GnRH derivatives for the assays were performed in cell culture medium (1% bacto tryptone and 0.1% yeast extract).

### 5.2. Model Cell

*Tetrahymena pyriformis* GL taxon was used in the logarithmic phase of growth. The cells were grown in culture medium with 1% *w*/*w* bacto tryptone (Sigma-Aldrich, St. Louis, MO, USA) and 0.1% *w*/*w* yeast extract (Difco Laboratories, Detroit, MI, USA) at 28 °C in atmospheric CO_2_/O_2_ ratio. The density of *Tetrahymena* cultures studied was 10^4^ cells/mL.

### 5.3. Study of Swimming Behavior

The tested concentrations of GnRH peptides were chosen on the basis of our previous chemotaxis experiments [31]. The cells were treated with GnRH derivatives at concentrations of their maximal chemotactic effects (GnRH-I: 10^−6^ M, GnRH-III: 10^−6^ M, Ac-SHDWKPG-NH_2_: 10^−6^ M, [GnRH-III(C)]_2_: 10^−11^ M, [GnRH-III(CGFLG)]_2_: 10^−6^ M). The swimming/chemokinetic behavior of cells was studied in an Axio-Observer invert microscope (Carl Zeiss Microscopy GmbH, Munich, Germany) by using AxioVision Rel 4.7.1. Software (Carl Zeiss Microscopy GmbH, Munich, Germany). The swimming behavior of cells was video recorded with the time-laps module (duration time: 5 s, picture-taking speed: 18 pictures/s) at a magnification of 5×. Before recording, the cells were treated with either test substance (the final concentration indicated above) or cell culture medium (in case of control) in a separate test tube. Right after the treatment, 50 µL of the sample was pipetted onto each microscopic slide and covered with a cover slip. On average, 25 cells were tracked per field of view for each recorded video, and 4 time-lapse videos were taken at random positions for each slide. The movement analysis was done by the cell tracker module of this software. The movement of each cell was tracked for 2 × 25 frames per each video, and data for individual cells were obtained at each frame (one data point per frame). The experiment with this setup was repeated twice on two consecutive days. The mean velocity of cells (normalized to the control) and the tortuosity of the swimming tracks (calculated as the ratio of the total distance travelled by the cell during a 25 frame analysis time to the straight distance between the first and last data points) were used to characterize the swimming behavior. All data gained were compared to the responsiveness of nontreated cells (control).

The reason for the short time (5 s) applied to take videos is that *Tetrahymena* is a relatively fast swimmer and will swim out of the field of view with longer observation times. Tracked cells were those remaining in the visual field during the analysis. To ensure this, 2 × 25 frame periods of the total recordings (one at the beginning and another one at the end of the video) were analyzed. Because of the light sensitivity of *Tetrahymena*, the application of a short recording time was also essentially needed.

### 5.4. Analysis of Intracellular Hormone Content

The effects of various GnRHs on the intracellular hormone content of *Tetrahymena* were measured by flow cytometry following indirect immunostaining. For the detection of intracellular T3, serotonin, histamine, epinephrine, and endorphin contents, the cells were stimulated by GnRH peptides at concentrations identical to those where their chemotactic character was detected [31] for 10 min in culture medium. The logical basis for using this relatively short incubation time comes from a previous study, where the most significant effect of the investigated hypothalamo/hypophyseal hormones (gonadotropins and thyrotropin) was observed after 10 min treatment [30].

After the treatment, the cells were immediately fixed with 4% paraformaldehyde solution (dissolved in PBS) for 5 min and then washed twice in wash buffer (1% bovine serum albumin (BSA); 20 mM Tris-HCl; 0.9% NaCl; 0.05% Nonidet NP-40; pH = 8.2; all components were purchased from Sigma-Aldrich, St. Louis, MO, USA). To block nonspecific binding of antibodies, the cells were treated with blocking buffer (1% BSA in PBS) for 30 min at room temperature. Aliquots from cell suspensions (50 μL) were transferred into tubes, and 50 μL primary antibody (rabbit, polyclonal anti-T3 [30], anti-serotonin, anti-histamine [54], and anti-β-endorphin [7] antibodies purchased from Sigma-Aldrich, St. Louis, MO, USA; or rabbit polyclonal anti-epinephrine antibody [55] purchased from ABCAM, Cambridge, UK), diluted 1:200 in antibody buffer (1% BSA; 20 mM Tris-HCl; 0.9% NaCl), was added, and the samples were incubated for 30 min in darkness at room temperature. Negative controls were carried out with 50 μL PBS containing 10 mg/mL BSA instead of primary antibody. After washing two times with wash buffer (1% BSA; 20 mM Tris-HCl; 0.9% NaCl) to remove excess primary antibody, the cells were incubated with the fluorescein isothiocyanate (FITC)-labeled secondary antibody (anti-rabbit IgG purchased from Sigma-Aldrich, St. Louis, MO, USA) diluted 1:50 with PBS for 30 min at room temperature.

For controlling the specificity, the autofluorescence of the cells and nonspecificity of the secondary antibody were checked. Using this simplified strategy to evaluate the immunoreactivity, instead of testing the cross-reactivity of the antibodies, is supported by the following reasons. In the last two decades, several works focused on the detection of different intracellular hormones and on verification of antibodies for intracellular immunostaining by confocal microscopy in different model systems, including *Tetrahymena*. In these studies, different anti-human hormone antibodies were used to detect the presence, the amount, and the localization of amino acid-, peptide-, or protein-type hormones. For example, the intracellular localizations of epinephrine [56], serotonin [22,54,57], histamine [22,54,57] β-endorphin [57], and T3 [57] were verified by immunocytochemical methods followed by confocal microscopic observation. These confocal microscopic images were in good agreement with the quantitative fluorescence data provided by flow cytometry.

Measurements were carried out in a FACSCalibur flow cytometer (Becton-Dickinson, San Jose, CA, USA) using 10,000 cells for each measurement. The CellQuest Pro program (version 5.1, Becton-Dickinson, San Jose, CA, USA) was used to measure and analyze the data. For each antibody, the experiment was repeated twice within one or two weeks with using five parallels per treatment group.

### 5.5. Study of Cell Growth

The growth of *Tetrahymena* treated with different GnRH peptides between 10^−12^ and 10^−6^ M was determined by CASY TT (ACEA Biosciences, San Diego, CA, USA) after 6 and 24 h incubation. The starting cell concentration was 10^4^ cells/mL. The control group was treated with the equivalent volume of cell culture medium.

Based on the non-invasive, dye-free electrical current exclusion principle of CASY TT, the cell count and the viability of *Tetrahymena* samples were characterized. The main setting parameters allowed us to analyze 3 × 400 μL aliquots (100 μL cell suspension diluted in 5mL CASYton) by using a capillary pore size of 150 μm. Each measurement (one GnRH derivative in one concentration) was carried out four times, and there were two replicates for the whole experiment. CASYexcell 2.3 (ACEA Biosciences, San Diego, CA, USA) was used for data acquisition and evaluation. The change in the number of viable cells was normalized to the identical control, and this value was given as the “proliferation index” in percent (Prolif. ind.).

### 5.6. Statistical Analysis

Data generated with the CellQuest Pro or AxioVision Rel 4.7.1 software were exported to Excel 2010, and additional evaluation of data was done by OriginPro 9.0 (OriginLab Corporation, Northampton, MA, USA). Data shown in the figures and tables represent averages and ±SD values.

Statistical analysis was performed by using a one-way ANOVA coupled with F-tests (OriginPro 9.0) in the case of chemokinesis and proliferation measurements. Histograms provided by flow cytometry and CASY TT were further analyzed by the Kolmogorov-Smirnov test (CellQuest Pro and OriginPro 9.0). In the text of the manuscript, more detailed statistical data (degrees of freedom and *F*-value of F-statistics as well as the *p*-value) are specified for justifying the significance of the results; however, in the figures and tables, because of the lack of space, the significance levels are indicated in the following way: * *p* < 0.05; ** *p* < 0.01; and *** *p* < 0.001.

## Figures and Tables

**Figure 1 ijms-20-05711-f001:**
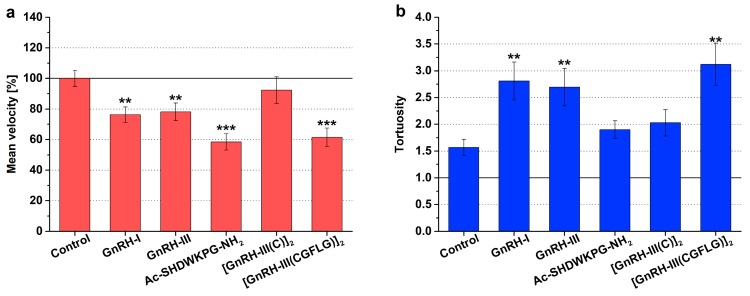
Effects of GnRH derivatives on the chemokinetic activity (mean velocity—**a**; tortuosity—**b**) of *Tetrahymena*. The effects of the GnRH peptides were studied in the following concentrations—GnRH-I: 10^−6^ M, GnRH-III: 10^−6^ M, Ac-SHDWKPG-NH_2_: 10^−6^ M, [GnRH-III(C)]_2_: 10^−11^ M, and [GnRH-III(CGFLG)]_2_: 10^−6^ M. The mean velocity is expressed as a percentage of the control. Data represent the mean of 4 parallels ± SD. The levels of significance are shown as follows: **, *p* < 0.01; ***, *p* < 0.001.

**Figure 2 ijms-20-05711-f002:**
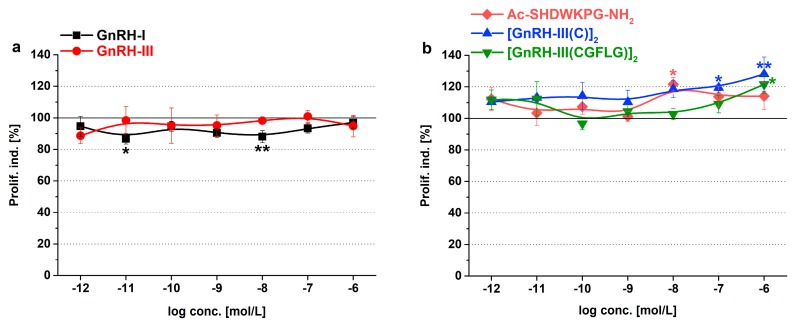
Short-term (6 h) effects of the GnRH derivatives on *Tetrahymena* proliferation. Proliferation index (Prolif. ind.) describes the number of viable cells normalized to control. Data represent the mean of 4 parallels ± SD. The levels of significance are shown as follows: *, *p* < 0.05; **, *p* < 0.01.

**Table 1 ijms-20-05711-t001:** Molecular composition of the tested gonadotropin-releasing hormone (GnRH) derivatives.

Groups	Peptide Names	Structures
Natural hormones	GnRH-I	<EHWSYGLRPG-NH_2_
	GnRH-III	<EHWSHDWKPG-NH_2_
GnRH-III fragment		Ac-SHDWKPG-NH_2_
Symmetric GnRH-III dimers	[GnRH-III(C)]_2_	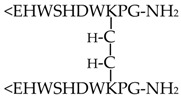
	[GnRH-III(CGFLG)]_2_	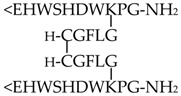

**Table 2 ijms-20-05711-t002:** Changes in the intracellular hormone contents induced by various GnRH peptides in *Tetrahymena*.

GnRH Peptides	Intracellular Hormone Content Mean Fluorescence Intensity (Geometric Mean Channel Value)				
	T3	Histamine	Serotonin	Epinephrine	Endorphin
**Control**	59.00 ± 1.28	21.02 ± 1.31	24.04 ± 0.37	100.82 ± 3.56	47.89 ± 1.26
**GnRH-I**	58.96 ± 2.01	24.89 *** ± 0.51	24.87 ± 0.59	108.83 ± 2.11	55.78 ** ± 0.51
**GnRH-III**	63.23 * ± 0.83	21.68 ± 0.92	19.65 ** ± 0.97	104.04 ± 1.87	61.52 *** ± 1.31
**Ac-SHDWKPG-NH** _**2**_	62.31 ± 1.94	30.27 *** ± 0.51	21.91 * ± 0.77	73.03 *** ± 1.34	43.85 ± 1.1
**[GnRH-III(C)]** _**2**_	65.08 ± 5.02	28.59 ** ± 0.93	22.01 * ± 0.58	78.44 ** ± 2.14	54.60 * ± 2.51
**[GnRH-III(CGFLG)]** _**2**_	54.56 * ± 0.55	27.93 * ± 1.75	23.01 ± 0.13	71.58 ** ± 2.11	47.45 ± 1.61

The effects of the GnRH peptides were studied in the following concentrations—GnRH-I: 10^−6^ M, GnRH-III: 10^−6^ M, Ac-SHDWKPG-NH_2_: 10^−6^ M, [GnRH-III(C)]_2_: 10^−11^ M, and [GnRH-III(CGFLG)]_2_: 10^−6^ M. The mean fluorescence intensity refers to the dimensionless geometric mean channel value. Data represent the mean of 5 parallels ± SD. The levels of significance are shown as follows: *, *p* < 0.07; **, *p* < 0.01; and ***, *p* < 0.001.

**Table 3 ijms-20-05711-t003:** Summary and comparison of chemosensory reactions induced by GnRH derivatives in *Tetrahymena*.

GnRH Peptides	Chemotactic Effects [31]	Chemokinetic Effects		Changes in Intracellular Hormone Contents					Cell Proliferation Modulator Effects	
		Mean velocity	Tortuosity	T3 rep ***/attr ** [12]	Histamine attr *** [12]	Serotonin rep *** [12]	Epinephrine neut ^a^	Endorphin attr * [2]	6 h	24 h
GnRH-I	attr *	-	++	0	+	0	0	+	-	0
GnRH-III	rep *	-	++	+	0	-	0	++	0	+
Ac-SHDWKPG-NH_2_	attr **	--	0	0	++	-	--	-	+	0
[GnRH-III(C)]_2_	attr **	0	0	0	++	-	--	+	++	0
[GnRH-III(CGFLG)]_2_	rep **	--	+++	-	++	0	--	0	+	0

attr, chemoattractant; rep, chemorepellent; +, ++, +++, increase in the investigated chemosensory reactions in different extent; -, -- decrease in the investigated chemosensory reactions in different extent; and 0, neutral effect. The level of significance is shown as follows: *, *p* < 0.07; **, *p* < 0.01; and ***, *p* < 0.001; ^a^ Unpublished results of the authors

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
