# Peer review of "Effects of Gonadotropin-Releasing Hormone (GnRH) and Its Analogues on the Physiological Behaviors and Hormone Content of Tetrahymena pyriformis"

_ijms, 2019, doi:10.3390/ijms20225711_

Round 1

Reviewer 1 Report

This is an interesting study, detailing the effects if various GnRH peptide fragments on Tetrahymena pyriformis motility, proliferation and hormone expression. The authors present a large amount of data, testing multiple different fragments in a range of assays. The paper, although a little verbose, is generally well written (although some modest editing of the written English is required). I have a few specific queries, and some minor suggestions, for the authors to consider.

1) Is there a difference in pH between GnRH fragments? Or is the difference in behaviour due to the structural differences between the peptides activating alternative/additional receptors?

2) Have the authors demonstrated GnRH-like signalling properties of these fragments, in a cell culture system i.e. can these peptides activate the GnRH-R from other species, as demonstrated through calcium mobilization etc.?

Figure 1 - have dose-response curves been constructed for these ligands? It is difficult to interpret the response to logarithmically different concentrations of peptides.

Table 2 - It is a little bit misleading to label these findings as intracellular hormone content; at best, these data show immunoreactivity. Do the authors have any cross-reactivity data for these antibodies? Do you have microscopy images of the immunostaining that might add specificity of the immunoreactivity?

Minor comments:

L45 ‘Chemotaxis’

L49 ‘highly sensible’ - what do the authors mean by this?

L54 Suggest ‘previous’ rather than ‘former’

L101 - ‘The results of’ - should this be ‘Structural homology of’?

L123 onwards; I am not familiar with formatting for this journal, but is it usual to provide numerical lists of results? If not, these sections should be edited to remove these numbered bullet points.

Author Response

Is there a difference in pH between GnRH fragments?

All compounds have very similar charge character. All of them have one positively charged functional group/unit (one for monomers and two for dimers) that are either on the α-amino or on the ε-amino group or on guanidine function of Arg. These functional groups are in TFA salt after HPLC purification and lyophilization. Therefore, the pH of the peptide solution in water is between 5 and 6 depending on the peptide concentration. However, the pH is buffered in the matrix used for the in vitro studies. Altogether, there is no significant difference in pH of the solution of compounds used for the experiments that can have any influence on it.

Or is the difference in behaviour due to the structural differences between the peptides activating alternative/additional receptors?

GnRH derivatives are very specific hormone peptides to the GnRH receptors. According to our knowledge, there is not any literature data showing the binding affinity of GnRH peptides to other receptors. For the specific receptor binding, the N- and C-terminal parts of the peptide are necessary. That might be the explanation for the opposite biological activity of the N-terminal truncated version than the full-length GnRH-III. Dimerized derivatives of GnRH-III retained the receptor binding affinity [1]. The dimer derivatives with two accessible binding sites show higher enzyme stability and may cross-link GnRH receptors but not different receptor types.

The GnRH receptors can be grouped into three distinct classes: type I, II and III. There are usually two to three forms of GnRH receptors present in most vertebrates [2]. The ligand selectivity, the binding affinity for different hormones and the function of these receptors are different. The type III GnRH receptor does not occur in mammals, while the type II GnRH receptor is inactivated or deleted in most mammals. It has arisen that the universally conserved type I GnRH receptor can take over the role of the type II receptor. The GnRH-II (and GnRH-III) has a low affinity to the type I receptor, and induces distinctly different signaling, while type II receptor is highly selective for GnRH-II [3, 4]. The expression of GnRH receptors in Tetrahymena has not been analyzed yet, but the potential presence of at least two types of GnRH receptors might be a possible explanation for the various cell biological behavior of GnRH derivatives. Millar and co-workers have suggested the ligand induced selective-signaling theory to clarify the different effects of the structurally different GnRH peptides. According to this theory, the different GnRH derivatives are able to preferentially stabilize different conformations of the same GnRH receptor, which generate distinct binding affinity and signaling output [5].

Have the authors demonstrated GnRH-like signalling properties of these fragments, in a cell culture system i.e. can these peptides activate the GnRH-R from other species, as demonstrated through calcium mobilization etc.?

In our previous study, we investigated the chemotactic signaling of different GnRH derivatives by determining the activation of phospholipase C (PLC) and/or phosphatidylinositol 3-kinase (PI3K) in Tetrahymena pyriformis ciliate protozoon and Mono Mac 6 human monocytic cell line. The flow cytometric measurement revealed that the chemoattractant GnRH peptides could not induce PLC activation on either of the model cells. By using PI3K inhibitors (wortmannin and LY294002) we could demonstrate that the GnRH derivatives exerted their chemoattractant effects, at least in part, via a PI3K-dependent mechanism in both model cells [6]. These findings indicate that putative GnRH receptors expressed on both model cells might be functionally different from pituitary receptors which are coupled with Gαq/11-PLC pathway, and behave more like the receptors in some tumor cells, where the PI3K was shown to implicate in the modulator effects of GnRH analogs on apoptosis and invasion. Since these results indicated a GnRH induced signal transduction pathway independent of PLC activation and consequently cytosolic Ca2+ release, we did not investigate the intracellular calcium level during that study.

Herédi-Szabó and co-workers reported that the secondary messenger pathway activated by GnRH-III proved to be different depending on cell type and location. GnRH-III decreased the cAMP level in human colon (HT-29) and pancreatic (PANC-1) cells via activation of Gi pathway, while in case of breast cancer cells (MDA-MB 231 and MCF-7) an increased intracellular [Ca2+] was observed [7].

In a superfused rat pituitary cell system, we could also show that the GnRH-III and its dimers retained a weak hormonal activity, which also suggests that these peptides could operate a signaling mechanism leading to LH secretion [1].

Figure 1 - have dose-response curves been constructed for these ligands? It is difficult to interpret the response to logarithmically different concentrations of peptides.

In the case of the investigation of chemokinesis and hormonal content, there were no dose-response curves constructed. In the present work, the five, tested GnRH derivatives and their concentrations were chosen based on our previous study about the chemotactic activity of the natural analogs and different GnRH-III derivatives tested in a concentration range 10-12-10-6 M [6]. The concentrations, that were applied in the measurements of chemokinesis – swimming behavior and hormone content, were identical to those where the most significant chemotactic character (chemoattractant: GnRH-I at 10-6 M, Ac-SHDWKPG-NH2 at 10-6 M, [GnRH-III(C)]2 at 10-11 M; chemorepellent: GnRH-III at 10-6 M, [GnRH-III(CGFLG)]2 at 10-6 M) was previously observed. In the present work we make an attempt to evaluate the complex cell biological activity of GnRH peptides and to find some interplays among their behaviors. Thus, it was reasonable to use the same concentrations, those were found to be the most active ones in the preliminary chemotaxis assay, during the present experiments, too.

Most of these aspects were included in the manuscript.

Page 3

In the present work, five GnRH derivatives (GnRH-I at 10-6 M, GnRH-III at 10-6 M, Ac-SHDWKPG-NH2 at 10-6 M, [GnRH-III(C)]2 at 10-11 M and [GnRH-III(CGFLG)]2 at 10-6 M), that were found chemotactically active in our previous study [31], were investigated. The results of the assays with GnRH peptides are categorized and presented in the following groups way: (i) natural hormones (GnRH-I and GnRH-III), (ii) N-terminal truncated fragment (Ac-SHDWKPG-NH2), (iii) symmetric GnRH-III dimers ([GnRH-III(C)]2 and [GnRH-III(CGFLG)]2). The molecular composition of the tested GnRH derivatives is shown in Table 1. The concentrations, that were applied in the measurements of chemokinesis – swimming behavior and hormone content, were identical to those where the most significant chemotactic character (chemoattractant: GnRH-I, Ac-SHDWKPG-NH2, [GnRH-III(C)]2; chemorepellent: GnRH-III, [GnRH-III(CGFLG)]2, was observed in our previous study. The different cell physiological factors of Tetrahymena, investigated in our present work, and the effects of the mammalian hormones including GnRHs on them are not completely independent from each other. In order to investigate the connection among the different cell physiological activities (e.g. chemotactic, chemokinetic and endocrine activities) induced by GnRH derivatives, it was reasonable to apply the same concentration (found the most active one in the chemotaxis assay) during the other measurements, as well.

Table 2 - It is a little bit misleading to label these findings as intracellular hormone content; at best, these data show immunoreactivity. Do the authors have any cross-reactivity data for these antibodies? Do you have microscopy images of the immunostaining that might add specificity of the immunoreactivity?

During the present work, we did not test the antibodies for cross-reactivity in order to check and verify their specificity.

The reason for using the simplified strategy of evaluating immunoreactivity is based on the following findings. In the last two decades, several works focused on the detection of different intracellular hormones and on verification of antibodies for intracellular immunostaining by confocal microscopy in different model systems including Tetrahymena. In these studies different anti-human hormone antibodies were used to detect the presence, the amount and the localization of amino acid-, peptide- or protein-type hormones.  For example, the intracellular localization of epinephrine [8], serotonin [9-11], histamine [9-11], β-endorphin [9] and triiodothyronine [9] was verified by immunocytochemical methods followed by confocal microscopic observation. These confocal microscopic images were in good agreement with the quantitative fluorescence data provided by flow cytometer.

The specificity of the anti-serotonin antibody used in most of the above-mentioned studies was also checked and verified by saturating the antibody with serotonin in mouse peritoneal cells. After the saturation with serotonin, the markedly decreased fluorescence detected by both flow cytometry and confocal microscopy was convincing [12].

Authors agree that the reasons for the simplified strategy should be mentioned even in the text, therefore a concise interpretation is inserted even to the relevant part of Materials and Methods

Page 12

For controlling the specificity, the autofluorescence of the cells and non-specificity of the secondary antibody were checked. Using this simplified strategy for evaluation of immunereactivity instead of testing the cross-reactivity of the antibodies is supported by the following reasons. In the last two decades, several works focused on the detection of different intracellular hormones and on verification of antibodies for intracellular immunostaining by confocal microscopy in different model systems including Tetrahymena. In these studies different anti-human hormone antibodies were used to detect the presence, the amount and the localization of amino acid-, peptide- or protein-type hormones.  For example, the intracellular localization of epinephrine [56], serotonin [57-59], histamine [57-59], β-endorphin [57] and triiodothyronine [57] was verified by immunocytochemical methods followed by confocal microscopic observation. These confocal microscopic images were in good agreement with the quantitative fluorescence data provided by flow cytometer.

Moderate English changes required.

… generally well written (although some modest editing of the written English is required).

Authors apologize for the grammar mistakes and they tried to do their best to minimize the grammatical inaccuracies. (A native English colleague has also provided proofreading as the minor changes show the improved parts of the text.)

L45: “The cChemotaxis is a directed movement…

L49: “Former data about the highly sensible sensitive chemotactic responsiveness…

L54: “…other hormones were studied in Tetrahymena in several former previous experiments.

L101: “The results of the assays with GnRH peptides are categorized and presented in the following groups way:...

I am not familiar with formatting for this journal, but is it usual to provide numerical lists of results? If not, these sections should be edited to remove these numbered bullet points.

Authors agree with the Referee’s suggestion and accordingly the text was edited to remove the numbered bullet points throughout the Results section.

References

MezÅ‘, G.; Czajlik, A.; Manea, M.; Jakab, A.; Farkas, V.; Majer, Z.; Vass, E.; Bodor, A.; Kapuvári, B.; Boldizsár, M.; Vincze, B.; Csuka, O.; Kovács, M.; Przybylski, M.; Perczel, A.; Hudecz, F., Structure, enzymatic stability and antitumor activity of sea lamprey GnRH-III and its dimer derivatives. Peptides 2007, 28, 806-820. Sower, S. A.; Decatur, W. A.; Joseph, N. T.; Freamat, M., Evolution of vertebrate GnRH receptors from the perspective of a Basal vertebrate. Front. Endocrinol. (Lausanne) 2012, 3, 140. Millar, R. P.; Lu, Z. L.; Pawson, A. J.; Flanagan, C. A.; Morgan, K.; Maudsley, S. R., Gonadotropin-releasing hormone receptors. Endocr. Rev. 2004, 25, 235-275. Schneider, F.; Tomek, W.; Grundker, C., Gonadotropin-releasing hormone (GnRH) and its natural analogues: A review. Theriogenology 2006, 66, 691-709. Millar, R. P.; Pawson, A. J.; Morgan, K.; Rissman, E. F.; Lu, Z. L., Diversity of actions of GnRHs mediated by ligand-induced selective signaling. Front. Neuroendocrinol. 2008, 29, 17-35. Lajkó, E.; Szabó, I.; Andódy, K.; Pungor, A.; MezÅ‘, G.; KÅ‘hidai, L., Investigation on chemotactic drug targeting (chemotaxis and adhesion) inducer effect of GnRH-III derivatives in Tetrahymena and human leukemia cell line. J. Pept. Sci. 2013, 19, 46-58. Herédi-Szabó, K.; Murphy, R. F.; Lovas, S., Different signal responses to Lamprey GnRH-III in human cancer cells. Int. J. Pept. Res. Ther. 2006, 12, 359-364. Csaba, G.; Kovacs, P.; Pallinger, E., Effects of different fixatives on demonstrating epinephrine and ACTH hormones in Tetrahymena. Biotech Histochem 2009, 84, 261-265. Csaba, G.; Kovacs, P.; Pallinger, E., Increased hormone levels in Tetrahymena after long-lasting starvation. Cell Biol Int 2007, 31, 924-928. Csaba, G.; Kovacs, P.; Pallinger, E., How does the unicellular Tetrahymena utilise the hormones that it produces? Paying a visit to the realm of atto-and zeptomolar concentrations. Cell Tissue Res 2007, 327, 199-203. Csaba, G.; Kovacs, P.; Pallinger, E., EDAC fixation increases the demonstrability of biogenic amines in the unicellular Tetrahymena: a flow cytometric and confocal microscopic comparative analysis. Cell Biol Int 2006, 30, 345-348. Csaba, G.; Kovacs, P.; Buzas, E.; Mazan, M.; Pallinger, E., Serotonin content is elevated in the immune cells of histidine decarboxylase gene knock-out (HDCKO) mice. Focus on mast cells. Inflamm Res 2007, 56, 89-92.

Reviewer 2 Report

This is an interesting paper in which the authors examine the impact of GnRH peptides on behavioral and neurotransmitter expression in the tetrahymena pyriformis.  Specifically, they measured the effect of the GnRHs on chemokinetic activity, proliferation and on intracellular hormone content.  While there is much merit to this study and potential significance of the data, there are gaps in the manuscript that should be addressed:

Broadly, the study utilizes a single cell organism as model to understand the impact of GnRHs on function.  There is no mention in the manuscript as to how much GnRHs is released.  For example, in the current assay of 10,000 cells per mL. how much GnRH is present in the media after a window of time?  How does this compare to the micro-molar concentration used?  What is the half life of these GnRHs?   The GnRH-III fragment data is very interesting because this, in particular, provides support for the biological importance of peptide fragments.  To start with, is this fragment found in this organism?  This needs to be mentioned or some rationale for using this fragment is needed.  To address biological importance, in more complex mammals, the endopeptidase (THOP1) cleaves GnRHI to produce fragments that have alternate biological activity.  Others include IGF-I, angiotensin, vasopressin.  The current study provides evidence that this principle is conserved through evolution.  Are there similar peptidases that cleave GnRHs in this species?  If so, what is their activity?   Figure 1.  It is unclear if a time course was conducted.  The method for this experiment is vague and needs to be clarified.  If a time course was not conducted, one is needed.  Also, a video describing the analysis used is needed.  How is tortuosity measured?  What are the controls (positive and negative controls?)? Tables 2 and 3. It is unclear if a time course was conducted.  It is also unclear how long after treatment were the cells fixed.  If a time course was not conducted, it should be done.  The technique utilized in this study requires more controls - eg, antibody specificity (see https://elifesciences.org/articles/48363).  How were these commercial antibodies verified for tetrahymena?  It is unclear what is actually measured since an antibody may not actually measure the same peptide/neurotransmitter across species.   In the measure of intracellular hormone content, how is the relative levels normalized?  Since FACS sorting is used, is cell number consistent?   How does GnRHs affect secreted hormones including GnRH? Statistics.  It is unclear how many replicates were conducted for all the experiments. Were the replicates conducted on different days or all conducted on the same day. Statistics. The references to statistics should include F statistics.  Results.  This section is very densely written and should be better organized for clarity.

Author Response

There is no mention in the manuscript as to how much GnRHs is released.  For example, in the current assay of 10,000 cells per mL. How much GnRH is present in the media after a window of time?  How does this compare to the micro-molar concentration used?

The actual manuscript refers to data on the effects of GnRH and its derivatives on intracellular hormone contents of Tetrahymena. The assayed hormones were triiodothyronine (T3), histamine, serotonin, epinephrine and endorphin. No endogenous or released GnRH was measured in this study.

What is the half life of these GnRHs?

The stability of GnRH peptides has not been studied in the cell culturing medium (containing Bacto-tryptone and yeast extract) of Tetrahymena, so far. Our previous results demonstrating the higher stability of GnRH-III dimers against hydrolytic enzymes (e.g. α-chymotrypsin) than the monomer GnRH-III suggest that the half-life of the dimer derivatives in the experimental condition used in our current work is also higher than that of the GnRH-III [1].

The GnRH-III and its analogs have been extensively investigated in our laboratory for application in targeted chemotherapy as a part of drug delivery systems. To evaluate the suitability of drug-containing GnRH-III derivative conjugates for tumor-selective drug-targeting, their stability/degradation were studied in different system, e.g. in human serum. Most of the conjugates proved to be stable at least for 24 h in human serum since only the intact conjugates were able to be detected with liquid chromatography in combination with mass spectrometry [13, 14]. However, in these studies the GnRH-III or its derivatives were covalently linked to a chemotherapeutic agent, but these studies support that the GnRH-III peptide variants themselves would be stable in the cell culture or conditional medium or Tetrahymena at least for the short time period (10-15 min) used in the different assays of our current work.

…is this fragment found in this organism? This needs to be mentioned or some rationale for using this fragment is needed.

According to our knowledge, GnRH hormones or any of their fragments (e.g. Ac-SHDWKPG-NH2) have not been identified in Tetrahymena pyriformis yet.

Although, the GnRH peptides are relatively sensitive to different endo- and exopeptidase (e.g α-chymotrypsin [1], angiotensin-converting enzyme and thimet oligopeptidase [15]), only a few papers are available about the biological or medical significance of analogs with shortened GnRH peptide chain. Some preliminary works were carried out in the 1970s and 1980s with the N- or C-terminal truncated GnRH-I fragments [16, 17] or hexapeptide GnRH-I analogs [18] and a wide range of binding affinities and biological responses were shown in those studies. Wu’s group have recently shown that GnRH-(1-5), produced by thimet oligopeptidase (zinc metalloendopeptidase EC3.4.24.15) has biological activity (e.g. regulating GnRH-I synthesis, secretion and migration of GnRH neurons), which could be different or even antagonistic compared to the parent peptide [15, 19].

Previously, the stability of GnRH-III and its dimer derivatives was analysed against α-chymotrypsin in our laboratory. The main cleavage site of this enzyme was found in the peptide bond between Trp3 and Ser4 amino acid residues, and the major fragments identified after 1 h reaction time was the H-4SHDWKPG10-NH2 [1]. This fragment and some other N-terminal truncated GnRH-III peptides were analysed in our laboratory. In spite of the importance of the conserved N-terminal sequence for receptor binding, the cellular uptake of these shortened GnRH-III peptides proved to be similar to the full-length GnRH-III in HT-29 human colon and MCF-7 human breast cancer cells [20]. Moreover, these fragments were found to be active in Tetrahymena and monocytes; their effects on chemotaxis and adhesion demonstrated the significance of the length and physicochemical character of the N-terminal amino acid of the GnRH-III fragments [6].

Based on these findings it was also grounded to investigate the GnRH-III fragments, Ac-SHDWKPG-NH2 – which has not been characterized yet – on the chemosensory functions of Tetrahymena and compare its effects with that of the full-length GnRH-III.

The manuscript was completed with a description of why the GnRH-III fragment was investigated.

Page 2-3

This fragment and some other N-terminal truncated GnRH-III peptides were previously analysed in our laboratory. In spite of the importance of the conserved N-terminal sequence for receptor binding, the cellular uptake of shortened GnRH-III peptides proved to be similar to the full-length GnRH-III in HT-29 human colon and MCF-7 human breast cancer cells [32]. Although, shorter GnRH sequences are frequently produced as a results of normal proteolytic degradation by different endo- and exopeptidase (e.g α-chymotrypsin [33], angiotensin-converting enzyme and thimet oligopeptidase [34]), only few preliminary works were carried out in the 1970s and 1980s with the N- or C-terminal truncated  GnRH-I fragments [35, 36] or hexapeptide GnRH-I analogs [37] and a wide range of binding affinities (equal to or lower than that of the GnRH) and biological responses (both agonistic and antagonistic) were shown in those studies. Wu’s group have recently shown that GnRH-(1-5), produced by thimet oligopeptidase (zinc metalloendopeptidase EC3.4.24.15) has biological activity (e.g. regulating GnRH-I synthesis, secretion and migration of GnRH neurons), which could be different or even antagonistic compared to the parent peptide [34, 38].

To address biological importance, in more complex mammals, the endopeptidase (THOP1) cleaves GnRHI to produce fragments that have alternate biological activity. Others include IGF-I, angiotensin, vasopressin. The current study provides evidence that this principle is conserved through evolution. Are there similar peptidases that cleave GnRHs in this species? If so, what is their activity?

In the 1970s, there were already several reports on Tetrahymena protease activities. Later works revealed some information concerning the biochemical and structural characteristics of these activities [21, 22]. Different cysteine proteases, such as calpain [23] and cathepsins [22, 24] were identified inside Tetrahymena and/or in the cell culture medium. The protease activity of the cell culture medium is attributed to a secretory process which releases a high amount of lysosomal proteases. One of the secreted Tetrahymena cysteine proteases was designated as tetrain, and was proved to be as a member of cathepsin L subfamily [22]. A higher amount of tetrain was observed in the culture medium in the stationary phase than in the logarithmic phase, and tetrain has high activities at neutral to alkaline pH values, which is typical for the medium in stationary phase [21, 22].

Comparison of amino acid sequences deduced from the cDNA sequence of cysteine protease genes of Tetrahymena thermophila to those of known cysteine protease revealed two distinct subfamilies within the cysteine proteases: proteins with highly conserved ERFNIN motif and the cathepsin B-like enzymes, that lack this motif [24].

The calpains, the other group of cysteine proteases were also identified in the cytosol of Tetrahymena. Based on the genome sequence of Tetrahymena thermophila large number of calpain-like proteins were predicted. The substrates of Tetrahymena calpains have not been fully known, but it is likely that numerous membrane-bound or membrane-associated proteins belong to their target [23].

There is no available data whether the Tetrahymena cysteine proteases could cleave any peptide-type or protein-type hormone produced and secreted by Tetrahymena. Based on our previous study dealing with the degradation of different drug delivery conjugates containing GnRH-III or its dimer derivatives in rat liver lysosomal homogenate [13, 14], it is assumed that the lysosomal enzymes secreted from Tetrahymena might be able to cleave the exogenously given GnRH hormones or any other pituitary-like material produced in Tetrahymena.

Figure 1.  It is unclear if a time course was conducted.  The method for this experiment is vague and needs to be clarified.  If a time course was not conducted, one is needed.

During the experiments on chemokinesis – swimming behavior, not the effects of the pretreatments with different GnRH derivatives but their acute effects were investigated. Therefore, we did not conduct any time course measurement.

The description of the method used to measure the swimming behavior of Tetrahymena was clarified with some additional information and explanation as you can see below.

Page 11-12

The swimming tracksbehavior of cells were video recorded registered with the Time-laps module (duration time: 5 sec, picture taking speed: 18 pictures/sec) at a magnification of 5x. Before the recording, the cells were treated with either test substance (the final concentration indicated above) or cell culture medium (in case of control) in a separate test tube. Right after the treatment, 50 µl of sample was pipetted onto each microscopic slide and covered with a coverslip. On average 25 cells were tracked per field of view for each recorded video, and 4 time lapse videos were taken at random position for each slide. The movement analysis was done by the Cell tracker module of this software. The movement of each cells were tracked for 2 x 25 frames per each video. data for individual cells were obtained at each frame (one data point per frame). The experiment with this setup was repeated twice on two consecutive days. Characteristics of tracking analysis were 25 cells/visual field, four parallel fields and 2x25 frames long analysis time. The mean velocity of cells (normalized to the control) and the tortuosity of the swimming tracks (calculated as the ratio of the total distance travelled by the cell during 25 frames long analysis time and the straight distance between the first and last data points) the ratio of the distance of starting and endpoint of the path and the real length of the swimming path) were used to characterize the swimming behavior. All data gained were compared to the responsiveness of non-treated cells (control).

The reason for the short time (5 s) applied to take videos is that the Tetrahymena is a relatively fast swimming unicellular and it swims easily out the field of view in case of the longer observation time. Tracked cells were those remaining in the visual filed during the analysis. To ensure this, 2 x 25 frames long periods of the total recordings (one at the beginning and other one at the end of the video) were analyzed. Because of the photosensitivity of the Tetrahymena, the application of the short recording time was also essentially needed.

Also, a video describing the analysis used is needed.  How is tortuosity measured?  What are the controls (positive and negative controls?)?

The analysis of the swimming behavior was carried by the in-built algorithm of the tracking module. By this tracking module the identification of cells, tracking the path of individual cells and the calculation of the different indices of the movement were automatically carried out. The automatic cell identification was needed to be adjusted in order to exclude cells, those were on the edge of visual field or swam out of it during 25 frames long analysis time. In case of each video, the tracking step was also checked and when the automatic tracking was not able to follow the cell swimming (however, the cell was still visible) it was manually corrected. In our opinion a short (25 frame long) video without detailed explanation would not be informative and taking a video to present the tracking analysis in detail is beyond the scope of this article.

The tortuosity parameter was determined from the recorded position data and provided by the above mentioned tracking module. The tortuosity is calculated as the ratio of the total distance travelled by the cell through 25 frames and the straight distance between the first and last data points of the actual path. “If the value of tortuosity is 1 it means that the track of the swimming is straight; if this value is between 1 and 10 the cell swims in a more spiral (winding) path, and if the tortuosity is greater than 10 Tetrahymena performs a slow circular type of movement known as creeping [13]” (Figure 1).

Figure 1 Swimming pattern of Tetrahymena categorized on the basis of tortupsity (please see the attachment)

Cellular sample treated with adequate volume of pure cell culture medium served as the negative control. In this experiment, there was no positive control applied.

Tables 2 and 3. It is unclear if a time course was conducted.  It is also unclear how long after treatment were the cells fixed.  If a time course was not conducted, it should be done.

In the case of the investigation of hormonal content, there were no time course conducted. The cells were stimulated with the different GnRH derivatives for 10 min, in concentrations identical to those where their chemotactic character was previously detected [6]. After the 10 min treatment, the cells were immediately fixed with 4% paraformaldehyde solution (dissolved in PBS).

The usage of 10 min incubation time was based on a previous study investigating the regulatory effect of gonadotropins (FSH and LH) and thyrotropin (TSH) on the triiodothyronine production of Tetrahymena. The cells were treated with these tropic hormones for 10, 20, 30 and 60 min. The TSH was shown to enhance the T3 content up to 30 min incubation time, but the most significant increase was detected after 10 min. The gonadotropins also increased the intracellular T3 content; however, only at 10 min. We chose this 10 min incubation in our present work for the sake of comparison of these two studies about the endocrine effects of different hypothalamo-hypophyseal hormones on Tetrahymena.

According to this answer the methods part of the manuscript was completed.

Page 12

For the detection of the intracellular hormone – T3, serotonin, histamine, epinephrine and endorphin – contents the cells were stimulated by GnRH peptides in concentrations identical to those where their chemotactic character was detected [31] for 10 minutes in culture medium. The logical basis for using this relatively short incubation time is a previous study, where the most significant effect of the investigated hypothalamo-hypophyseal hormones (gonadotropins and thyrotropin) was observed after 10 minutes’ treatment [53].

After the treatment, the cells were immediately fixed with 4% paraformaldehyde solution (dissolved in PBS) for 5 min...

The technique utilized in this study requires more controls - eg, antibody specificity (see https://elifesciences.org/articles/48363). How were these commercial antibodies verified for tetrahymena?  It is unclear what is actually measured since an antibody may not actually measure the same peptide/neurotransmitter across species.

Referee 1 had a very similar question regarding the verification of the specificity of antibodies used in the measurement of intracellular hormone content. The answer given to his/her question is copied below with some modifications.

During the present work, we did not test the antibodies for cross-reactivity in order to check and verify their specificity. Authors understand the need for testing the antibodies for cross-reactivity and for a proof that they are specific for Tetrahymeny hormone. Using a simplified strategy of evaluation immunoreactivity is supported by the following reasons. In the last two decades, several works focused on the detection of different intracellular hormones and on verification of antibodies for intracellular immunostaining by confocal microscopy in different model systems including Tetrahymena. In these studies, different anti-human hormone antibodies were used to detect the presence, the amount and the localization of amino acid-, peptide- or protein-type hormones. For example, the intracellular localization of epinephrine [8], serotonin [9-11], histamine [9-11], β-endorphin [9] and triiodothyronine [9] was verified by immunocytochemical methods followed by confocal microscopic observation. These confocal microscopic images were in good agreement with the quantitative fluorescence data provided by flow cytometer.

The specificity of the anti-serotonin antibody used in most of the above-mentioned studies was also checked and verified by saturating the antibody with serotonin in mouse peritoneal cells. After the saturation with serotonin, the markedly decreased fluorescence detected by both flow cytometry and confocal microscopy was convincing [12]. In case of the insulin, the specificity of an immunoassay, that was used for its identification, was confirmed by isolating the substance designated as insulin and testing its bioactivity, which proved to be homologous to the mammalian hormone [25].

Authors agree that the reasons for the simplified strategy should be mentioned even in the text, therefore a concise interpretation is inserted even to the relevant part of Materials and Methods.

Page 12

For controlling the specificity, the autofluorescence of the cells and non-specificity of the secondary antibody were checked. Using this simplified strategy for evaluation of immunoreactivity instead of testing the cross-reactivity of the antibodies is supported by the following reasons. In the last two decades, several works focused on the detection of different intracellular hormones and on verification of antibodies for intracellular immunostaining by confocal microscopy in different model systems including Tetrahymena. In these studies different anti-human hormone antibodies were used to detect the presence, the amount and the localization of amino acid-, peptide- or protein-type hormones.  For example, the intracellular localization of epinephrine [56], serotonin [57-59], histamine [57-59], β-endorphin [57] and triiodothyronine [57] was verified by immunocytochemical methods followed by confocal microscopic observation. These confocal microscopic images were in good agreement with the quantitative fluorescence data provided by flow cytometer.

In the measure of intracellular hormone content, how is the relative levels normalized?  Since FACS sorting is used, is cell number consistent?

The geometric mean fluorescence intensity (GMFI) or geometric mean channel value (GeoMean), that was used in our study, correlates with the number of antibodies that recognize and attach accordingly to the cell antigens (hormones), thus allowing quantification of antigen expression (hormone content) per cell. By using an unstained control (where only secondary antibody was added to the sample) the level of the background fluorescence was determined, and consequently the voltage and gates were set appropriately. The GMFI obtained for each treatment group was normalized to that of the control group containing only secondary antibody. As it was written in the manuscript (page 12), 10000 cells were collected during the flow cytometric measurements. “The measurement was carried out in a FACSCalibur flow cytometer (Becton-Dickinson, San Jose, CA, USA), using 10000 cells for each measurement.”

How does GnRHs affect secreted hormones including GnRH?

As it was written above the actual manuscript refers data on effects of GnRH and its derivatives on intracellular hormone (T3, histamine, serotonin, epinephrine and endorphin) contents of Tetrahymena. No endogenous or secreted GnRH was measured in this study. However, it is an interesting question whether GnRHs affect the secretion of hormones produced inside the Tetrahymena. This problem was also arisen in the Authors and it has been already mentioned in the Discussion part. This problem is now further discussed in the manuscript.

Page 7-8

The fact that intracellular hormome content of Tetrahymena could be regulated by external hormonal stimuli might predict that the cells could respond to this kind of stimulations by secreting these endogenous substances. The intracellular hormones investigated in our experiment are known to have significant chemotactic [2, 12] and therefore chemokinetic activities. This means that after their release these hormones could react to the same cells’ (autocrine) and/or to other cells’ (paracrine) physiological activities [3]. The autocrine/paracrine effect of the intracellular hormones might indirectly affect the migratory response of the cells to exogenously GnRH peptides. The changes in the intracellular hormone contents could be followed by the altered secretion of them, which might indirectly affect the migratory response of the cells to an exogenously given hormone (e.g. GnRH) by autocrine/paracrine manner. For example, we hypothesize that the secretion of serotonin, a strong chemorepellent hormone, might modify the chemokinetic effect of the exogeneously given GnRHs by provoking a slower and more winding swimming pattern. The opposite, a fast, more linear swimming might be caused by secreting the histamine or endorphin (both of them are chemoattractant).

For the more accurate evaluation of the paracrine/autocrine regulatory function of GnRH derivatives, further investigations – detection of secreted endogenous hormones outside the cells – would be needed. To study the concentrations and activity of the hormones released from the cells some factors should be taken into account: (i) whether these hormones are secreted a constitutive or regulated way, (ii) what ratio of the induced endogenous hormone can be secreted; (iii) the results of modulation of different intracellular hormone contents might have positive or negative effect on the release of each other; (iviii) molecular interactions (e.g. enhancing or neutralizing) between environmental substances and the released endogenous hormones and (v) the optimal time frame and concentration range of the paracrine/autocrine hormones. The complexity of the problem is increased further by considering the chemokinetic/chemotactic reactions of Tetrahymena. The constant swimming behavior of ciliates consecutively destroys the gradients of secreted hormones around the cells. Therefore, it is supposed that we have to count with more homogenous hormone level(s) which elicits more chemokinetic than chemotactic responses.”

It is worth also considering that a minimal cell density is required for reliable analytical detection. There is a unique relation among cell density, concentrations of hormones released and distances between individual cells in the culture. If the number of the cells present in a culture is more, greater is the amount of the secreted hormone and less is the distance between the cells; however, the cells can be more destructive on the concentration gradient of the endogenous hormones.

It is unclear how many replicates were conducted for all the experiments. Were the replicates conducted on different days or all conducted on the same day.

Authors apologize for leaving out accidentally this important information from the Methods. The manuscript has been completed with the number of the parallels and replicates used in case of each measurement.

Page 11

On average 25 cells were tracked per field of view for each recorded video, and 4 time lapse videos were taken at random position for each slide. The movement analysis was done by the Cell tracker module of this software. The movement of each cells were tracked for 2 x 25 frames per each video. data for individual cells were obtained at each frame (one data point per frame). The experiment with this setup was repeated twice on two consecutive days.

Page 12

For each antibody, the experiment was repeated twice within one or two weeks with using five parallels per treatment group.

Page 13

Each measurement (one GnRH derivative in one concentration) was carried out four times and there were two replicates for the whole experiment.”

The references to statistics should include F statistics.

Authors agree with Referee 2’s suggestion and the Statistical analysis paragraph was modified accordingly.

Page 13

Statistical analysis was performed by using the one-way ANOVA coupled with F-testalgorithm (OriginPro 9.0)”

Results.  This section is very densely written and should be better organized for clarity.

Considering this suggestion and Referee 1’s comment, the Results section was shortened and edited to remove the bullet points in order to organize better the text and make it easier to understand.

Please find these modifications in the manuscript.

References:

13. Manea, M.; Leurs, U.; Orbán, E.; Baranyai, Z.; Ohlschlager, P.; Marquardt, A.; Schulcz, A.; Tejeda, M.; Kapuvári, B.; Továri, J.; MezÅ‘, G., Enhanced enzymatic stability and antitumor activity of daunorubicin-GnRH-III bioconjugates modified in position 4. Bioconjug. Chem. 2011, 22, 1320-1329.

Schreier, V. N.; Mezo, G.; Orban, E.; Durr, C.; Marquardt, A.; Manea, M., Synthesis, enzymatic stability and in vitro cytostatic effect of Daunorubicin-GnRH-III derivative dimers. Bioorg. Med. Chem. Lett. 2013, 23, 2145-2150. Cleverly, K.; Wu, T. J., Is the metalloendopeptidase EC 3.4.24.15 (EP24.15), the enzyme that cleaves luteinizing hormone-releasing hormone (LHRH), an activating enzyme? Reproduction 2010, 139, 319-330. Rivier, J.; Amoss, M.; Rivier, C.; Vale, W., Synthetic luteinizing hormone releasing factor. Short chain analogs. J. Med. Chem. 1974, 17, 230-233. Rivier, J.; Vale, W.; Burgus, R.; Ling, N.; Amoss, M.; Blackwell, R.; Guillemin, R., Synthetic luteinizing hormone-releasing factor analogs. Series of short-chain amide LRF homologs converging to the amino terminus. J. Med. Chem. 1973, 16, 545-549. Haviv, F.; Palabrica, C. A.; Bush, E. N.; Diaz, G.; Johnson, E. S.; Love, S.; Greer, J., Active reduced-size hexapeptide analogues of luteinizing hormone-releasing hormone. J Med Chem 1989, 32, 2340-2344. Larco, D. O.; Cho-Clark, M.; Mani, S. K.; Wu, T. J., The metabolite GnRH-(1-5) inhibits the migration of immortalized GnRH neurons. Endocrinology 2013, 154, 783-795. Szabo, I. Synthesis of GnRH-III derivatives to enhance its antitumor activity. Eötvös L. University, Budapest, 2009. Suzuki, K. M.; Hayashi, N.; Hosoya, N.; Takahashi, T.; Kosaka, T.; Hosoya, H., Secretion of tetrain, a Tetrahymena cysteine protease, as a mature enzyme and its identification as a member of the cathepsin L subfamily. Eur. J. Biochem. 1998, 254, 6-13. Suzuki, K. M.; Hosoya, N.; Takahashi, T.; Kosaka, T.; Hosoya, H., Release of a newly-identified cysteine protease, tetrain, from Tetrahymena into culture medium during the cell growth. J. Biochem. 1997, 121, 642-647. Croall, D. E.; Ersfeld, K., The calpains: modular designs and functional diversity. Genome Biol 2007, 8, 218. Karrer, K. M.; Peiffer, S. L.; DiTomas, M. E., Two distinct gene subfamilies within the family of cysteine protease genes. Proc. Natl. Acad. Sci. U S A 1993, 90, 3063-3067. Le Roith, D.; Shiloach, J.; Heffron, R.; Rubinovitz, C.; Tanenbaum, R.; Roth, J., Insulin-related material in microbes: similarities and differences from mammalian insulins. Can. J. Biochem. Cell Biol. 1985, 63, 839-849.

Round 2

Reviewer 1 Report

Thank your for addressing my previous concerns; the revised manuscript reflects these appropriate alterations.

Author Response

Response #2 to review report #1 of Manuscript

Effects of gonadotropin-releasing hormone (GnRH) and its analogues on the physiological behaviors and hormone content of Tetrahymena pyriformis”.
 Manuscript ID: ijms-622300

We appreciate the Reviewer #1’s comments and most importantly the time She/He has spent on judging our manuscript.

Sincerely,

Dr Laszlo Kohidai

Reviewer 2 Report

The authors have addressed some of the concerns of the authors.  There are some issues that remain to be addressed:

F-statistics.  In the text (Results section), the authors should include with each p value, the df's, etc.  For example, (F2,38 =5.449, p<0.009). The lack of time course is disconcerting.  The authors need to better develop a rationale why the time course is not conducted.  For example, it should be sufficiently simple to video at 2 or more time points with treatment. Merely citing that the swimming behavior followed a software is not sufficient.  The criteria for movement and defining other parameters are needed.  As it is, it would be impossible to replicate this work in another laboratory without these information even with the same software.  
